# Ascension of Chlamydia is moderated by uterine peristalsis and the neutrophil response to infection

Torrington Callan[1], Stephen Woodcock[1]*, Wilhelmina May Huston[2]*

**1** Faculty of Science, School of Mathematical and Physical Sciences University of Technology Sydney, Sydney, Australia, **2** Faculty of Science, School of Life Sciences, University of Technology Sydney, Sydney, Australia

* Stephen.Woodcock@uts.edu.au (SW); Wilhelmina.Huston@uts.edu.au (WMH)

## Abstract

*Chlamydia trachomatis* is a common sexually transmitted infection that is associated with a range of serious reproductive tract sequelae including in women Pelvic Inflammatory Disease (PID), tubal factor infertility, and ectopic pregnancy. Ascension of the pathogen beyond the cervix and into the upper reproductive tract is thought to be necessary for these pathologies. However, *Chlamydia trachomatis* does not encode a mechanism for movement on its genome, and so the processes that facilitate ascension have not been elucidated. Here, we evaluate the factors that may influence chlamydial ascension in women. We constructed a mathematical model based on a set of stochastic dynamics to elucidate the moderating factors that might influence ascension of infections in the first month of an infection. In the simulations conducted from the stochastic model, 36% of infections ascended, but only 9% had more than 1000 bacteria ascend. The results of the simulations indicated that infectious load and the peristaltic contractions moderate ascension and are inter-related in impact. Smaller initial loads were much more likely to ascend. Ascension was found to be dependent on the neutrophil response. Overall, our results indicate that infectious load, menstrual cycle timing, and the neutrophil response are critical factors in chlamydial ascension in women.

**Data Availability Statement:** All relevant data are within the manuscript and its Supporting Information files.

## Author summary

Modelling informs elements of the disease process that are difficult to evaluate due to the complexity and rareness of these factors in human studies. Previous research has led to significant advances in understanding the role of the immune system and pathogen characteristics in the biology of *Chlamydia trachomatis*. Here we combine this body of knowledge to build a model that allows us to understand how and when *Chlamydia trachomatis* might progress to the upper reproductive tract in women. Elements of the natural reproductive cycle and immune system were found to be important in moderating the disease process in women.

**Funding:** This work is supported by an Australian Government Research Training Program (RTP) scholarship awarded to TC (https://www.dese.gov.au/research-block-grants/research-training-program). The funders had no role in study design, data collection and analysis, decision to publish, or preparation of the manuscript.

**Competing interests:** The authors have declared that no competing interests exist.

## Introduction

There is considerable evidence to indicate that *Chlamydia trachomatis* is one of the aetiological agents responsible for Pelvic Inflammatory Disease (PID), and that women who experience PID are more likely to experience tubal factor infertility and ectopic pregnancy [1–4]. Laparoscopic, histological and molecular evidence has indicated that infection can be present in the fallopian tubes, even when non-detectable at the cervix, the primary site of infection [5]. It is suggested that the development of tubal pathology is a result of a chlamydial infection ascending into the upper reproductive tract combined with a damaging pro-inflammatory response to the pathogen [6].

While the proportion of infections that ascend are not known, there are estimates on the frequency of damaging pathology. In the POPI trial, 9.5% women with a chlamydial infection developed PID [1]. A longitudinal study of 1844 women with laparoscopically confirmed PID (all aetiologies) found that approximately 30% experience tubal factor infertility [3]. Ascension of *Chlamydia trachomatis* into the upper genital tract can occur without necessarily developing pathology. The relatively low rate of pathology development in women with infections [7] suggests that a number of host and pathogen factors moderate ascension, such as menstrual cycle, stage of infection, immune status (and other influential factors not considered in this study, such as genotypic factors (host and pathogen), cellular exit strategies of the pathogen (lysis or extrusions) and host microbiome composition). The infectious load (the amount of infection when the infection is first received) may also determine the likelihood of ascension. Maxion et al. described how the infectious load of *Chlamydia muridarum* in mice was more likely to result in viable bacteria in the oviduct when the animals received lower infecting doses [8]. Higher infecting loads appeared to result in an immune profile in favour of cells associated with the innate immune response. This study found the response of immune cells (innate and adaptive) was independent of location (lower or upper genital tract) for the relationship with infecting load. The findings from this study suggested that innate immune cells contribute to the control of ascending infection, such that the immune response for high infectious loads of bacteria was sufficient to prevent ascension into the upper genital tract, but not so for lower loads.

A comprehensive review identified an association between infectious load, specimen type, and anatomical site of infection [9]. However, the evidence was unclear for the role of infectious load in ascending infection. An experiment on the effects of infectious load in female mice found that higher inoculating doses ascended to the upper genital tract more rapidly [10]. However, in the experimental protocol, mice were given 2.5g of medroxyprogesterone acetate 7 days before infection, and so the interplay between initial load and timing of the menstrual cycle was not evaluated. Only one study [11] looked at the relationship between load at the cervix and PID diagnosis, which found an association between higher load and PID. However, the study design was cross sectional, and so a causal interpretation cannot be made.

All *chlamydiae* share a unique biphasic developmental cycle, where infectious extracellular elementary bodies (EBs) enter cells and differentiate into replicative reticulate bodies (RBs). After several rounds of replication, RBs differentiate back into EBs and are released from the host cell, in order to infected neighbouring cells [12]. Each host (humans, guinea pig, mice etc.) generally has a distinct species of *Chlamydia*, and there are 12 species currently identified but the overall biphasic developmental cycle process is consistent throughout the genus. The *Chlamydiae* do not appear to encode a specific mechanism for movement on its genome. A one-dimensional spatiotemporal model described the population dynamics of chlamydial particles, infected and uninfected epithelial cells, and with movement that was determined by a diffusion coefficient in an idealised representation of the cervix, with random diffusion at

about 10–4 μm2 s-1 (with an uncertainty of an order of magnitude) in the elementary body form (EB) [13, 14]. The model demonstrated that the peak infection load and time to clearance was a function of the movement of *Chlamydia* (*C.*) *trachomatis*, and that increased motility may lead to stronger host immune response. However, the scale of the diffusion term, in comparison to the length of the cervix implies that diffusion of the elementary bodies is an implausible process for ascension.

A model for movement or ascension has been proposed by which polymorphonuclear leukocytes (PMNs) infiltrated the conjunctival epithelium following a guinea pig infection with *Chlamydia caviae* [15]. The response was in the early stages of the infection, which concords with the previously described critical role of neutrophils in controlling the early stage of an infection [16]. The PMN response was observed to result in detached superficial epithelia from the mucosa, including infected cells and elementary bodies that remained healthy and intact during the process. It was suggested by Rank et al. that this response was a defensive mechanism from the perspective of the host organism in that it allows for infected cells that have been ejected from the epithelium to be cleared from the infection site. Further, it was proposed that from the perspective of the pathogen, this process allowed for movement away from infected sites where there is a lack of susceptible uninfected cells, transporting to new infection sites via fluid dynamics or tissue movement. Movement and secretion of materials via fluid dynamics in the genital tract is governed by uterine peristalsis. The interplay of peristaltic contractions and ascension of sexually transmitted infections has not been examined and represents a crucial gap in current understanding.

It has been established that there is bi-directionality of the uterine peristalsis dependent on the stage of the menstrual cycle. In work conducted Kunz and Leyendecker (2001), vaginal sonography of uterine peristalsis in women has been used to characterise the dynamics of rapid sperm transport [17]. They found an increase in the frequency and intensity of subendometrial and myometrial peristaltic waves during the follicular phase of the menstrual cycle. The number of waves in the fundo-cervical direction decreased and waves in the cervico-fundal direction increased at this stage of the cycle, suggesting that sperm transport is facilitated by these uterine conditions, and that the extent of migration is increased with the progression of the follicular phase.

It is therefore plausible that the immune response which detaches infected cells from the epithelium and into the cervical canal can result in the transport of infected cells and elementary bodies via peristalsis into higher regions of the genital tract. Since the direction and intensity of peristaltic movement is dependent on the menstrual cycle, the stage of the menstrual cycle at infection could be one governing factor in determining which women experience infections that ascend or descend the genital tract, although it may be further impacted by the infectious burden (as explored in this model)

The phenomena described above could be combined to provide a process for infective chlamydial bodies to enter into the genital tract and be transported to the upper reproductive tract via peristalsis. From there, the infection cycle described above could start, with pro-inflammatory immune responses following the infection sites to the upper genital tract. In this work, we represent this process with stochastic model of the dynamics of a chlamydial infection, the associated host response, and the movement of infectious material. This allows us to better appreciate the complex, dynamic, and nonlinear interactions between the neutrophil response, the infectious load, and the menstrual cycle that moderate chlamydial ascension. Elucidating the processes involved in chlamydial ascension will provide a better understanding of the pathogenesis of *C. trachomatis* associated PID and infertility. In turn, the moderating factors could enable more specific testing for at-risk women, or development of other interventions when infections are diagnosed.

## Methods

### Developing a stochastic model to determine the relative influence of factors that moderate ascension

We developed a stochastic model of *C. trachomatis* and infected cells in two compartments; the cervical epithelium and cervical canal. The structure of the model is informed by previous mathematical modelling studies of within-host chlamydial dynamics [13, 18], and by synthesis of the literature on the chlamydial developmental cycle and host cell interactions. The model developed for this study extends previous models in order to incorporate elements specific to ascension (such as peristalsis, neutrophil ejections, epithelial and cervical elements) and stochastic elements to incorporate uncertainty. Fig 1 displays the host cell interactions that we represent in our model in order to evaluate the moderating impact of each on ascension.

We infer parameters for our model from plausible prior distributions (Table 1). In turn, prior distributions are parameterised where possible, using values from previous modelling studies. Transitions are stochastic meaning realisations of the model permit variable outcomes.

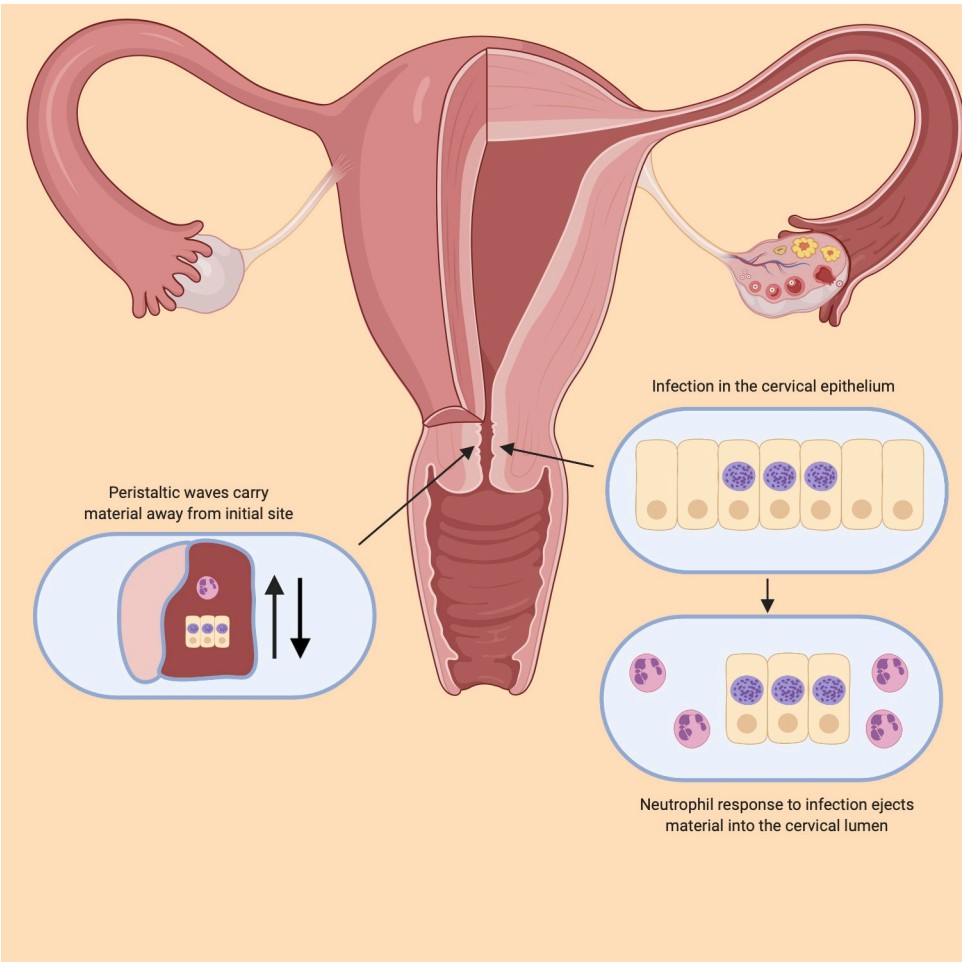

**Fig 1. The proposed model of chlamydial ascension.** The figure depicts the female reproductive tract and elements that were evaluated in their moderating impact of ascension. The factors evaluated were the chlamydial cell cycle, the host and pathogen interactions including the immune system response and the ejection of infectious material from the epithelium, and the effect of peristalsis in moving material either away or towards the upper reproductive tract depending on the stage of the menstrual cycle. Fig 1 was created with BioRender.com.

**Table 1. Parameters incorporated in the model and values assigned for this study.**

| Parameter | Meaning | Distribution* | Value | Supporting Literature |
|---|---|---|---|---|
| $\gamma_{1,innate}$, $\gamma_{1,adaptive}$ | Death rate of elementary bodies (EBs) (per minute) due to the innate and adaptive immune response | Truncated Gamma | 1, 100, $\tau_1$ | [13, 18, 22, 23] |
| $\gamma_{2,innate}$, $\gamma_{2,adaptive}$ | Death rate of infected cells (per minute) due to immune response | Truncated Gamma | 1, 100, $\tau_2$ | [13, 18, 22, 23] |
| $\tau_1$ | Threshold of EBs at which immune system increases | Negative Binomial | 5000, $\sigma_1$ | Assumed |
| $\tau_2$ | Threshold of infected cells at which immune system increases | Negative Binomial | 150, $\sigma_2$ | Assumed |
| $\sigma_1$ | Variance of the threshold parameter | Beta | 1, 1 | Assumed (hyperparameter) |
| $\sigma_2$ | Variances of the threshold parameter | Beta | 1, 1 | Assumed (hyperparameter) |
| $\kappa_1$ | Rate of inclusion of EBs (per minute) into new infected cells | Gamma | 1, 100 | [13, 18, 22, 23] |
| $\kappa_2$ | Rate of burst of infected cells (per minute) to new EBs | Gamma | 1, 100 | [13, 18, 22, 23] |
| $\alpha$ | Rate of migration of infectious material due to the neutrophil response | Gamma | 1, 100 | Informed by [15] |
| $I$ | Number of new inclusions of infected cells | Negative Binomial | 10, $\sigma_4$ | [13, 18, 22, 23] |
| $B$ | Number of new EBs in each burst event | Negative Binomial | 350, $\sigma_3$ | Mean and dispersion chosen to match the 200–500 nominal range as per [13, 18, 22, 23] |
| $M_1$ | Number of bacteria per migration event | Negative Binomial | 100, $\sigma_5$ | Informed by [15] |
| $M_2$ | Number of infected cells per migration event | Negative Binomial | 10, $\sigma_6$ | Informed by [15] |
| $\sigma_3$ | Variance of the inclusion size | Beta | 2, 2 | Mean and dispersion chosen to match the 200–500 nominal range as per [13, 18, 22, 23] |
| $\sigma_4$ | Variance of the burst size | Beta | 2, 2 | Assumed (hyperparameter) |
| $\sigma_5$ | Variance of the migration for EBs | Beta | 1, 1 | Assumed (hyperparameter) |
| $\sigma_6$ | Variances of the migration for infected cells | Beta | 1, 1 | Assumed (hyperparameter) |
| $\gamma_{3,fixed}$ | Rate of removal (per minute) of infectious material due to the intensity of peristaltic contractions | Gamma | 1, 1 | Assumed (unique to this study) |

*Details on the parameterisation used for each of the distributions in the model can be found in the S1 Text and S1 Fig.

This allows us to infer behaviour about an infection over a broad range of plausible parameter values, including phenomena not explicitly represented in the model.

Finally, the simulations are classified by a binary condition that reflects if the population of infectious material in the cervical canal compartment is non-zero at the endpoint of the simulation. If there is infectious material present (i.e. non-zero in the cervical canal), then ascension is said to occur in that iteration. This outcome is then used to determine relationships between the parameters of the process and the proportion of simulations where ascension occurred (reflecting peristalsis towards the fundus and live infectious material in the cervical canal). Relationships between the parameter values and the outcome are graphically summarised using natural splines. Each simulation is begun with a fixed amount of time remaining before peristaltic contractions move fluid in the cervical canal upwards, causing any infectious material to ascend. We stop each simulation at the moment when peristalsis causes material to begin ascending, with the assumption that any material in the cervical canal at this point will ascend.

## Defining the process

There are two entities of interest in this modelling exercise, extracellular chlamydial elementary bodies (EBs) and infected cells containing chlamydial reticulate bodies (RBs) and EBs [12]. We denote

$X_{\{C_1,t\}}, X_{\{C_2,t\}}, Y_{\{C_1,t\}}, Y_{\{C_2,t\}}$ as random variables denoting the number of extracellular chlamydia (represented as $X$) and infected cells (represented as $Y$) in the epithelium and cervical canal respectively (represented by the subscripts $C_1$, $C_2$). The model assumes that infected cells produce extracellular bacteria in an instantaneous burst, as opposed to a continuous process, which is important to note as this choice may impact the behaviour of generated quantities of the model [19].

*C. trachomatis* has two distinct exit strategies from a cell, lysis and extrusions, which are mediated by distinct mechanisms [20]. Extruded chlamydial inclusions may exit the cell slower than what occurs via lysis in the rate of exit, and the amount of pathogen released from the cell may differ (albeit this difference is likely subtle compared to the other parameters we explore here). Further, these factors governing chlamydial cell exit may vary by serovar type and various host specific factors. Therefore, we utilise broad parameter ranges over the function of extracellular *C. trachomatis* in order to represent the variation attributable to these factors without explicitly representing them. To this end, there is only one type of extracellular *C. trachomatis* represented in the model, although we recognise that there are many other potentially influential factors in the process.

We utilise the framework of previous models of chlamydial infections by Wilson to develop our model [18]. The realised outcomes of a particular sample path in a model of *C. trachomatis* is proposed to be strongly influenced by the random elements [21], hence our model incorporates stochasticity. Representing the process as a stochastic model allows us to represent the uncertainty attributable to other factors that are believed to be important in moderating infections, such a host and pathogen genotype and the cervico-vaginal microbiome.

For the ascension model, we extend the dynamics of the process to include ejection from the epithelia into the cervical canal. We denote with the subscript $C_1$ for infected cells and EBs in the epithelium, and $C_2$ for infectious units in the cervical canal. The scheme of the process can be represented as follows (following the convention of Pearson *et al.* [19].

$$X_{C_1} \to \emptyset \ rate \ \gamma_1 (\text{EBs per minute})$$

$$Y_{C_1} \to \emptyset \ rate \ \gamma_2 (\text{infected cells per minute})$$

$$I \, X_{C_1} \to Y_{C_1} \ rate \ \kappa_1 (\text{EBs per minute})$$

$$Y_{C_1} \to B \, X_{C_1} \ rate \ \kappa_2 (\text{infected cells per minute})$$

$$M_1 X_{C_1} \to M_1 X_{C_2} \ rate \ \alpha (\text{EBs per minute})$$

$$M_2 Y_{C_1} \to M_2 Y_{C_2} \ rate \ \alpha (\text{infected cells per minute})$$

$$X_{C_2} \to \emptyset \ rate \ \gamma_3 (\text{EBs per minute})$$

$$Y_{C_2} \to \emptyset \ rate \ \gamma_3 (\text{infected cells per minute})$$

$$Y_{C_2} \to B \, X_{C_2} \ rate \ \kappa_3 (\text{infected cells per minute})$$

The vector $(X_{\{C_1,t\}}, X_{\{C_2,t\}}, Y_{\{C_1,t\}}, Y_{\{C_2,t\}})$ represents the current state vector of the process. The empty set $\emptyset$ is used to denote an infectious body (EB or infectious cell) that has been removed by the immune system or peristaltic movement. For this model, we have possible

transitions that correspond to the removal of infectious material: (-1, 0, 0, 0) representing the death of an extracellular EB, (0, -1, 0, 0) representing the death of an epithelial infected cell, and (0, 0, -1, -1) representing the removal of EBs and infected cells in the cervical canal.

$I(t)$, $B(t)$ are random variables that determine the number of bacteria required to form an infectious unit, and the number of bacteria formed by a bursting event. $M_1(t)$, $M_2(t)$ are random variables describing the ejection from the epithelium to the cervical canal of EBs and infected cells respectively.

The trading between infectious material in the epithelium is characterised by $(-I(t), 1, 0, 0)$, where $I(t)$ EBs from an inclusion to produce one infected cell. Conversely, infected cells can burst to produce $B(t)$ new EBs, represented by $(B(t), -1, 0, 0)$. Ejection is represented by $(-M_1(t), -M_2(t), M_1(t), M_2(t))$. Infected cells in the cervical canal can burst and produce EBs, represented by $(0, 0, B(t), -1)$. This final transition does not explicitly represent extrusions, however the parameters that govern the function of extracellular *C. trachomatis* are allowed to vary over ranges that could include this form of *C. trachomatis*.

## Simulating the model

We simulate the model in continuous time, where time is measured in minutes. We use a grid of initial conditions for $X_{\{C_1, t=0\}} = x_0$ (from 1 to 10,001 in steps of 100, with 10 simulations at each value) and stage of the menstrual cycle (from 0 to 43,200 minutes in steps of 1800, with 10 simulations at each value) for a total of 252, 500 simulations, and assume that $X_{\{C_2, t=0\}} = Y_{\{C_1, t=0\}} = Y_{\{C_2, t=0\}} = 0$. The longest simulation was therefore 43, 200 minutes long. A simulation either ended with material ascended, 0 bacterial load or with some infection present in the epithelium but not in the cervical canal. Events occur according to a Poisson process, with rate parameter $\lambda_t$ being the sum of all rate parameters as described in the process scheme above. Fig 2 describes the transitions between states.

The parameters $\gamma_1$ and $\gamma_2$ that govern the immune response are made up of two components, one representing the innate immune system and one representing the adaptive immune system. The adaptive immune response is assumed to operate once a certain threshold level of each infectious unit is achieved and is zero otherwise. These two thresholds (one for EBs, one for infectious cells), are drawn from a negative binomial distribution, with mean 5000 for EBs and 150 for infected cells. The shape parameter for each threshold is drawn from a $Beta(1,1)$ distribution. It is assumed that the immune system response for extracellular bacteria is independent to the response for infected cells. The immune portion of the model is as follows

$$\gamma_1 = \gamma_{1,innate} + \gamma_{1,adaptive}$$

$$\gamma_2 = \gamma_{2,innate} + \gamma_{2,adaptive}$$

$$\gamma_{1,innate}, \gamma_{2,innate} \sim Gamma(1, 100)$$

$$\gamma_{1,adaptive} \sim \{0 \text{ if } X_{C_1} \leq \tau_1, Gamma(1, 100) \text{ if } X_{C_1} > \tau_1\}$$

$$\gamma_{2,adaptive} \sim \{0 \text{ if } Y_{C_1} \leq \tau_2, Gamma(1, 100) \text{ if } Y_{C_1} > \tau_2\}$$

$$\tau_1 \sim NegBinom(5000, \sigma_1)$$

$$\tau_2 \sim NegBinom(150, \sigma_2)$$

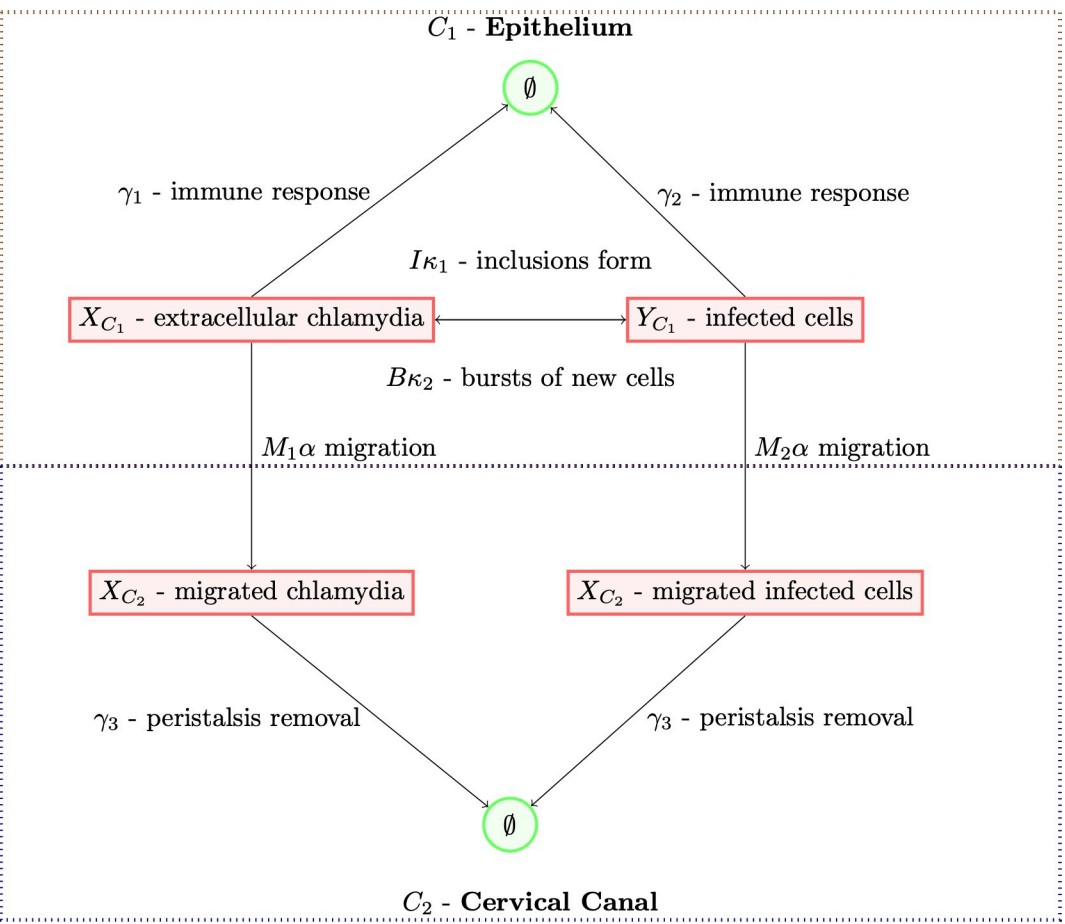

**Fig 2. Flow diagram of the model.** The figure outlines each component of the model that are incorporated in the analysis. Extracellular chlamydia can form inclusions to created new infected cells, and infected cells can burst to create new extracellular chlamydia. Both extracellular chlamydia and infected cells can be removed by the immune system, and be migrated to the cervical canal via the neutrophil response. In the cervical canal, infected cells can burst to create new chlamydia, but not vice versa. Extracellular Chlamydia and infected cells in the cervical canal be removed via peristalsis. The parameters of each component from the model are drawn from distributions that represent host and pathogen specific factors. The impact of each factor is represented as the proportion of simulations where the infection ascended for each parameter in the model in the following figures, out of a total of 252,500 simulations that were run.

$$\sigma_1, \sigma_2 \sim Beta(1,1)$$

A consideration for the model was the unique nature of chlamydial growth inside an intracellular inclusion vacuole where the organism replicates in multitudes, and the release of the infectious particles in bursts (by extrusion or lysis). The rate of inclusion developments and bursts are denoted by $\kappa_1$ and $\kappa_2$. The burst and inclusion sizes are drawn from a negative binomial with means 350 and 10 respectively, where the shape parameter comes from a $Beta(2,2)$ distribution. The chlamydial inclusion and burst portion of the model can be represented as follows

$$\kappa_1, \kappa_2 \sim Gamma(1,100)$$

$$B \sim NegBinom(350, \sigma_3)$$

$$I \sim NegBinom(10, \sigma_4)$$

$$\sigma_3, \sigma_4 \sim Beta(2, 2)$$

The rate of migration of infectious material is denoted as α, which assumes that the rate of migration is the same for both EBs and infected cells (i.e. representing cells displaced by the neutrophil response as outlined in the introduction). The number of EBs and cells being migrated due to the neutrophil response is also drawn from a negative binomial, with mean 100 and 10 respectively. The shape parameter for this random variable is drawn from a *Beta* (1,1) distribution. All shape parameters in the model act as hyperparameters for the uncertainties on the model parameters. They are rarely observed in the literature, and so they are drawn from weakly informative distributions about the exact parameter value. This addition of uncertainty makes the model results less sensitive to specific choices of parameter values.

$$\alpha \sim Gamma(1, 100)$$

$$M_1 \sim NegBinom(100, \sigma_5)$$

$$M_2 \sim NegBinom(10, \sigma_6)$$

$$\sigma_5, \sigma_6 \sim Beta(1, 1)$$

The parameter $\gamma_3(t)$ represents the removal rate of material in the cervical canal at time $t$. It is modelled as a fixed component and a function that represents variation due to the effect of the menstrual cycle on peristalsis. This assumes that changes in peristalsis occur continuously across the menstrual cycle, and that the menstrual cycle lasts for 30 days. 30 days was arbitrarily assigned, although the model parameters allow for a broad range of variability given the 21–35 day range in the population. The period and phase shift of the sine function are based on the number of minutes in 30 days, and the range is restricted to the interval [0, 1]. Peristaltic movement is represented as follows

$$\gamma_3(t) = \gamma_{3,fixed} + 0.5\, sin\left(\frac{\pi(t - 10800)}{21600}\right) + 0.5$$

$$\gamma_{3,fixed} \sim Gamma(1, 1)$$

The inclusion of probability distributions over the parameters governing the model can be thought of as including uncertainty of host and pathogen specific factors that are not otherwise explicitly modelled. Ascension is defined as having occurred within a simulation, if the population of extracellular *C.trachomatis* and infected cells, $X_{C_2}, Y_{C_2}$ is greater than zero when the simulation time finishes. We then summarise the relationship between parameter values of the process, and the proportion of simulations where ascension occurred. Relationships are summarised graphically in the results section using cubic splines.

## Results

### Modelling permits variation in ascension dynamics

Our modelling abstracts the ascension of *C. trachomatis* into a set of stochastic dynamics, that allows us to elucidate the moderating factors that might determine whether or not an infection ascends into the upper reproductive tract in women.

As outlined in the methods the factors include extracellular bacteria in the cervical epithelium infect cells which later burst and produce new amounts of bacteria. Immune responses, and in particular the presence of neutrophils, which eject material into the cervical lumen, and uterine peristalsis and the direction. Fig 3 indicates the factors that potentially determine chlamydial ascension. We evaluated three realizations of the same stochastic process with distinct parameter values, distinct initial conditions for infectious load and the same initial conditions

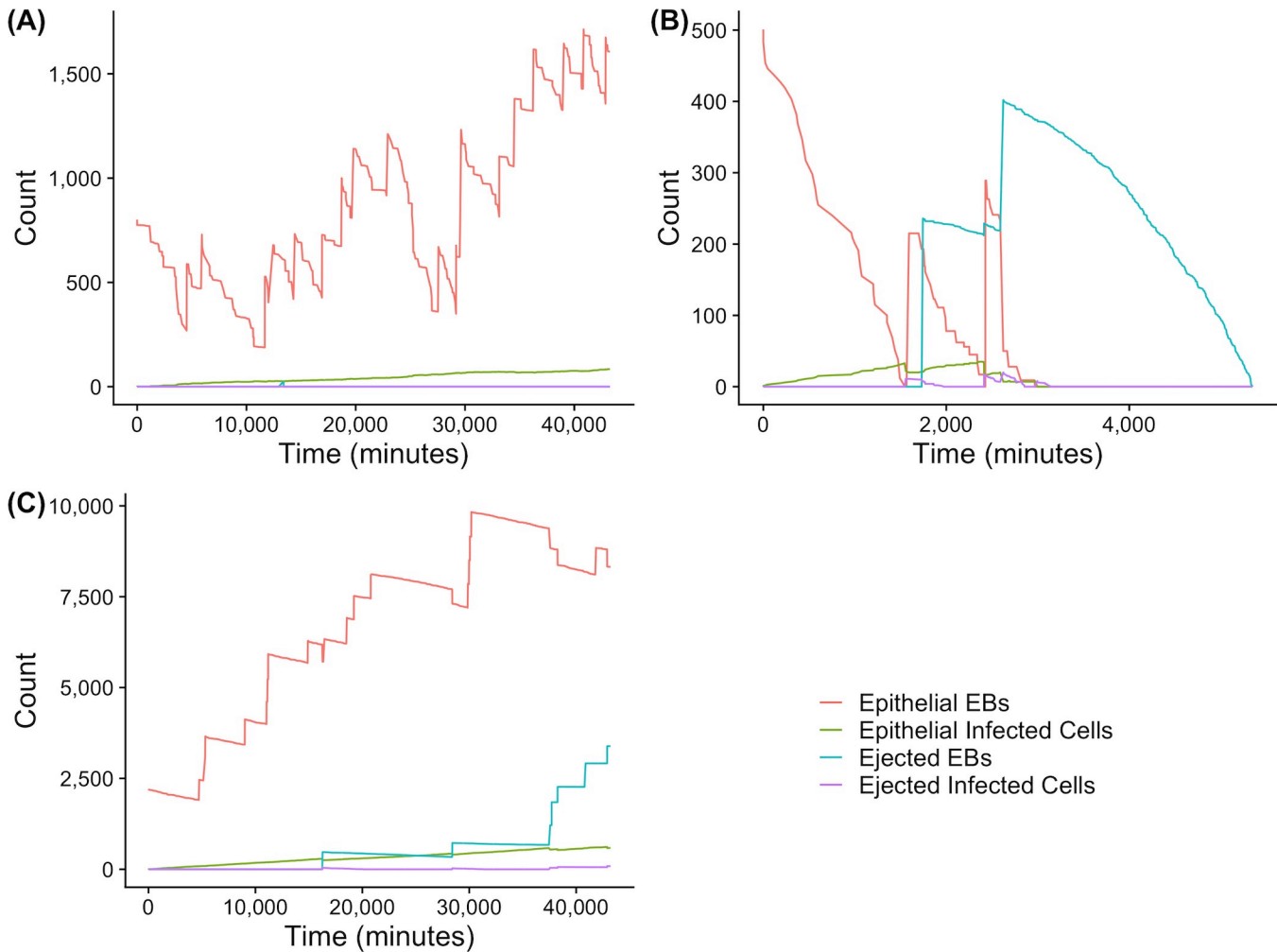

**Fig 3. Graphical representation of the counts for each status of elementary bodies and cells in the model under three distinct scenarios over time.** The figure shows the outputs of simulated sample paths of the model where factors have been modified. Each figure is a demonstration showing selected representations from the 252, 500 simulations that were conducted. (A) An example output from the model that starter with a load in the range of 801 EBs. (B) An example output of the model starting with a low initial load of 501 EBs. (C) An output that commended with a higher load than A and B of 2201 EBs. Each figure shows a graphical representation of count of each cell/chlamydia form represented on the y axis with the actual counts of *Chlamydia trachomatis* in the different forms indicated by the coloured lines (see figure legend bottom right), and time in minutes (x axis).

for time until cervico-fundal peristaltic contractions and the initial amounts of infected cells and ejected materials to show the major outcomes of the model (A, B, C, refer to the same plots in the figure) indicating the proportions of ascensions influenced by each set of parameters. In scenario A, the input is a high infectious load. In this scenario the infection grows rapidly over time, however insufficient material is ejected from the epithelium and the infection will not ascend (as indicated by the lack of ejected EBs in pale blue in Fig 3A). In scenario B, an output of the model where the input was a low infectious load, the infection is ejected at two distinct time points. However, this leaves insufficient material in the epithelium for the infection to replicate, and so the infection dies out there. Since the infection cannot replicate in the cervical canal, it to dies out well before peristalsis in the upwards direction begins. In the final example, scenario C, the infection is initiated with an even higher infectious load, and in this output from the model the infection survived in the epithelium whilst sufficient amounts of material are ejected (pale blue) for the infection to be present at the point where material can be transported into the upper genital tract.

Using the model we can examine the role of both burst rates (release of infectious *C. trachomatis* from cells) and inclusion formation rate (or our measure of establishing new inclusion vacuoles in new cells). This is based on the assumption that an increase in the frequency of bursts, would increase the number of new bacteria circulating in the epithelium. This is somewhat dependent on infectious dose, as higher infecting doses would have an increased frequency of bursts, but this factor reflects the infection across the course of time more generally. Increasing burst frequency does not appear to independently moderate the chances of ascension (Fig 4A). However, increasing the rate at which extracellular bacteria infect target cells does appear to increase the chances of ascension (Fig 4B). The chances of ascension when this rate was set at 0.001 infected cells per minute was estimated at 15.3%, 34.6% when the rate was 0.005 and 55.1 for a rate of 0.05.

Next, we examined simulations of the model to assess how frequently ascension occurs and the amount of bacteria that ascend. As Fig 5 demonstrates, even in infections where the bacteria ascend, there is a long tail in the distribution of how much bacteria will reach the upper genital tract. In our simulations, the median number of bacteria that ascended per infection was 0, with an interquartile range of 0 to 127. The top 25% of infections had between 127 and 59,090 bacteria ascended. This exercise gives an estimate of 36% of infections that ascend, but only 9% where more than 1000 bacteria managed to ascend, and 0.1% where more than 10,000 bacteria ascended.

## Modelling indicates a role for initial infectious load and timing relative to the menstrual cycle for ascension

Overall, initial infectious load was influential on ascension events. Specifically, in this model we found initial loads in the range of 1000–3500 were most likely to ascend, with an apparent decrease in ascension found for higher loads (Fig 6A). The timing of initial infection relative to the onset of peristaltic contractions in the cervico-fundal direction (which is moderated by the menstrual cycle) had a much more marked impact on the simulated chances of ascension (Fig 6B). For example, infections that were initiated 30 hours prior to the onset of upwards peristaltic movement had a 7.8% of ascension, whereas infections initiated 10 days or more prior had a 42% of ascension. The relationship between infectious load and timing was also found to be critical in influencing ascension. Infections with smaller infectious loads experience linearly decreasing chances of ascension as the menstrual cycle progresses, so an infection that occurs later in the cycle may have reduced chances of ascending. However, the opposite applies to high infectious loads, where infections have a higher chance of ascending as the menstrual

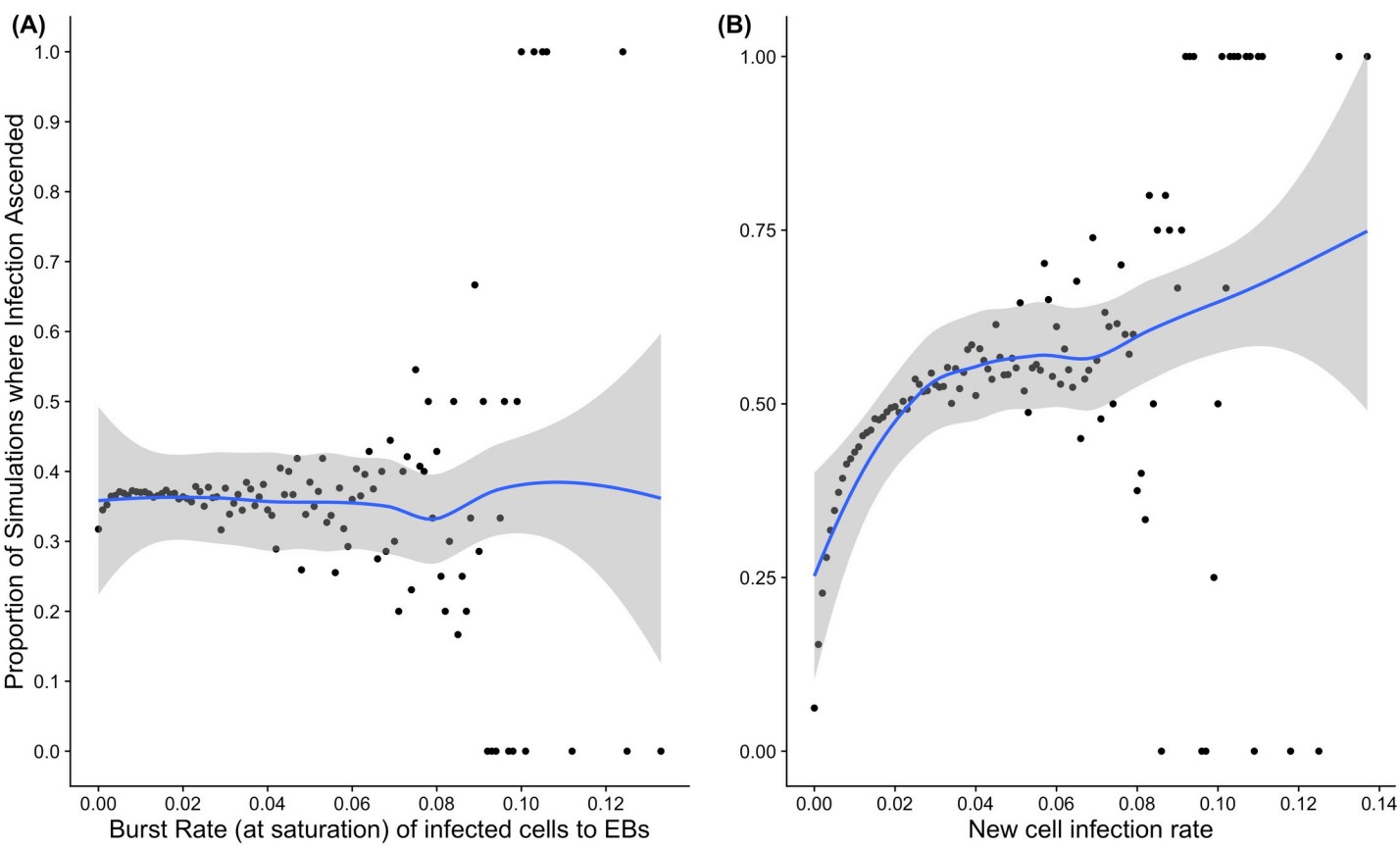

**Fig 4. Graphical representation of the proportion of ascension events in the simulations, as influenced by burst rate (A) and new cell infection rate (B)..** (A) The relationship between burst rate (in cells per minute) and the simulated chances of ascension. (B) The relationship between new cell infection rate (cells per minute). The y axis shows the proportion of simulations where infections ascended (out of 252, 500 simulations conducted), and x axis shows the different parameters values of burst rate and new cell infection rate. Each dot represents simulations at a particular parameter value. These are simulated over a range of infectious loads (10 simulations at each value of infectious load from 1 to 10001 in steps of 100. The blue line represents a loess fit of ascension proportion against each parameter value, where shading is the 95% confidence interval of the fit.

cycle progresses closer to peristaltic movement in the cervico-fundal direction (Fig 6C). Although a high initial load of 10001 had a broadly constant chance of ascension between 39–43%, for infections that were initiated 22 or more days prior to upwards peristaltic movement.

### Ascension chances are dependent on the neutrophil response in this model

Our simulation experiments show that the chances of ascension are broadly independent of the immune systems response (defined as overall scale of response, Fig 7), indicated by death rate of infectious material (extracellular EBs and infected cells) and moderated by the global immune response (separated into innate and adaptive systems). It is important to note that these phenomena represent the relative strength and weaknesses of the overall immune system and its responses between hosts. The model predicts that increasing or decreasing the strength of the host immune response, that determines the removal of EBs and infected cells, would have no impact on the chances of a particular infection ascending. The outcomes of the model under these parameters was that death of infected cells (Fig 7B and 7D) at a higher rate (rather than EBs, Fig 7A and 7C) was more likely to reduce the proportion of infections that ascended in the simulations (although the outcomes of the model at high death rates were highly variable 0–100% of ascensions).

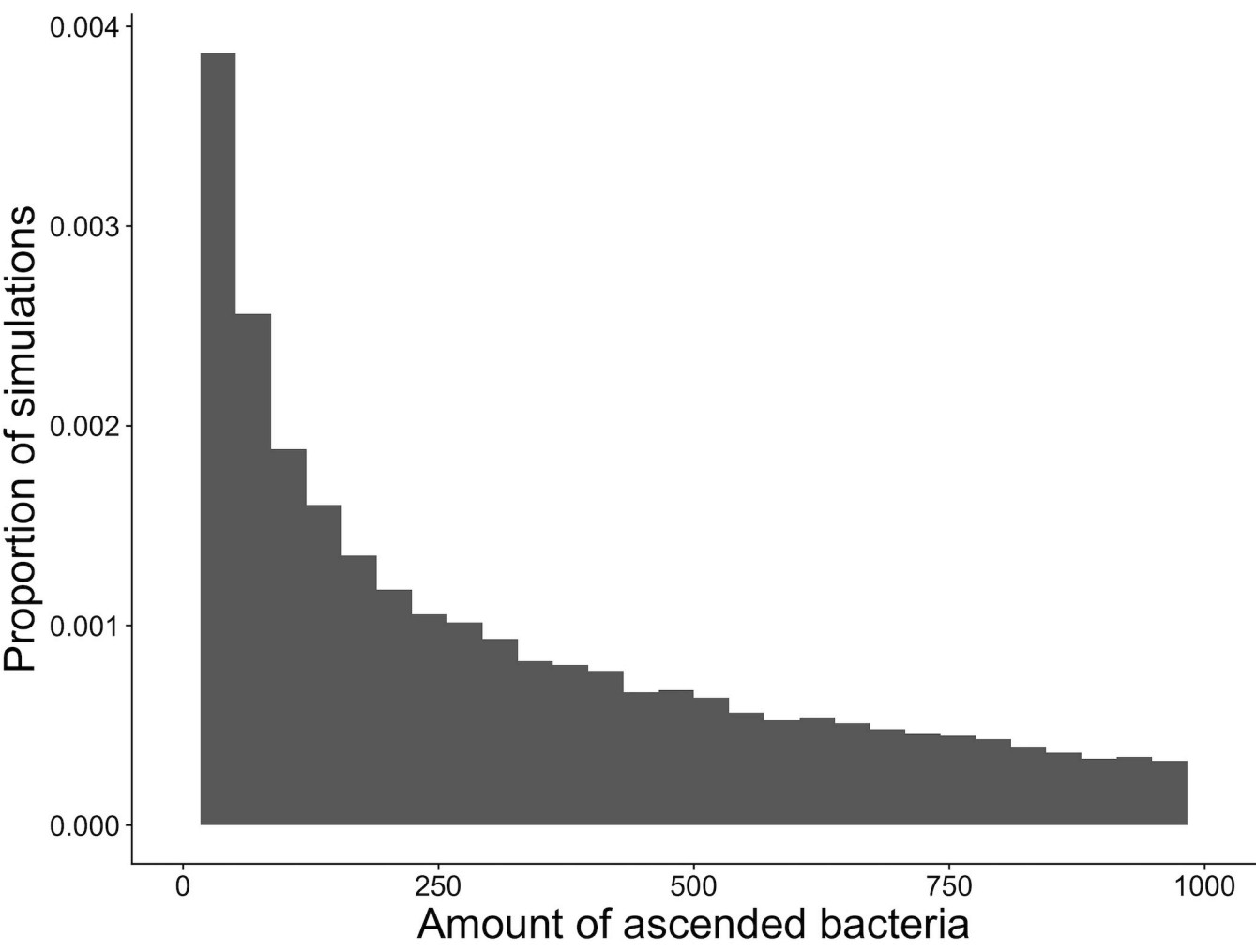

**Fig 5. Histogram of the proportion of simulations leading to ascension of bacteria, and the amount of bacteria predicted by the model to ascend.** The amount of bacteria predicted to ascend (x axis) and the proportions of each simulation that resulted in ascension (y axis) from 252, 500 simulations of the model. These are simulated over a range of infectious loads (10 simulations at each value of infectious load from 1 to 10001 in steps of 100.

Infections that instigate higher neutrophil responses are more likely to be one that ascend (Fig 8). The simulated probability of ascension was 15.8% for a rate of 0.001 (per minute), and 42.5% for a simulated rate of 0.01. The exact nature of this relationship depends on the stage of the menstrual cycle in which infection occurs. It is an assumption of the model that the neutrophil response and other features of the innate and adaptive immune response are related by their density-dependence. That is, the ejection rate of cells due to the neutrophil response is higher when the removal rate of infected cells due to the innate and adaptive response is higher, as both respond to the overall density of infected cells.

## Discussion

In this study we have applied a modelling approach to evaluate factors that influence chlamydial ascension in women. The impact of infectious burden on the likelihood of ascension shown as realisations in several iterations of the model presented here is not inconsistent with what might be expected from existing biological data in mouse and humans. It has been proposed

           

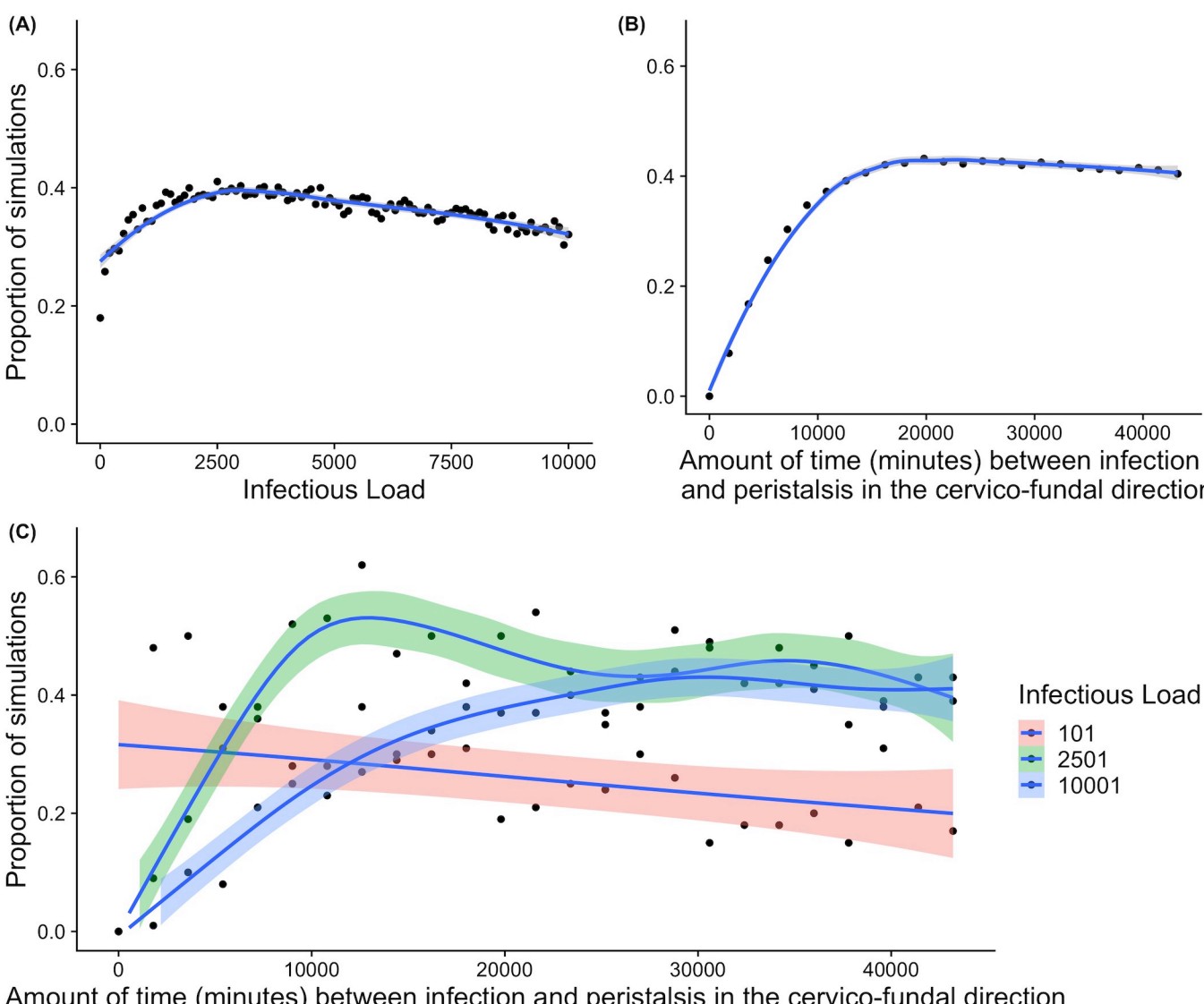

**Fig 6. Graphical summary of the proportions of simulations where ascension occurred and role of load (A), peristalsis (B), and both (C).** (A) Simulations analysing the impact of infectious load (the count of initial bacteria) on the the chances of ascension. (B) The time (in minutes) prior to the onset of peristaltic contractions in the cervico-fundal direction and chances of ascension. (C) The relationship between infectious load, timing of infection with respect to peristaltic contractions, and the chances of ascension. Each figure shows the proportion of simulations from the model where the infection ascended (proportion of representations from 252, 500 simulations) on the y axis, where the x axis represents the factor being modified (A. infectious load, B. peristalsis, C. peristalsis and grouped by infectious load). These are simulated over a range of infectious loads (10 simulations at each value of infectious load from 1 to 10001 in steps of 100. The blue lines represent a loess regression against the proportion of simulations resulting in ascension, and the shaded regions are 95% confidence intervals of the regression. Dots represent simulations at a particular parameter value.

that a strong adaptive immune system response may be sufficient to clear the infection from the cervix before ascension can occur [24]. Our modelling results indicate that ascension occurs independently of the strength of the immune system response, suggesting that the progression to pathological outcomes in the upper reproductive tract is not so straightforward. However, the presence of a neutrophil response has been previously associated with *C. trachomatis* and the development of pathology. There are a variety of studies that support the presence of neutrophils with *C. trachomatis* and endometritis [25, 26]. Implicating the neutrophil

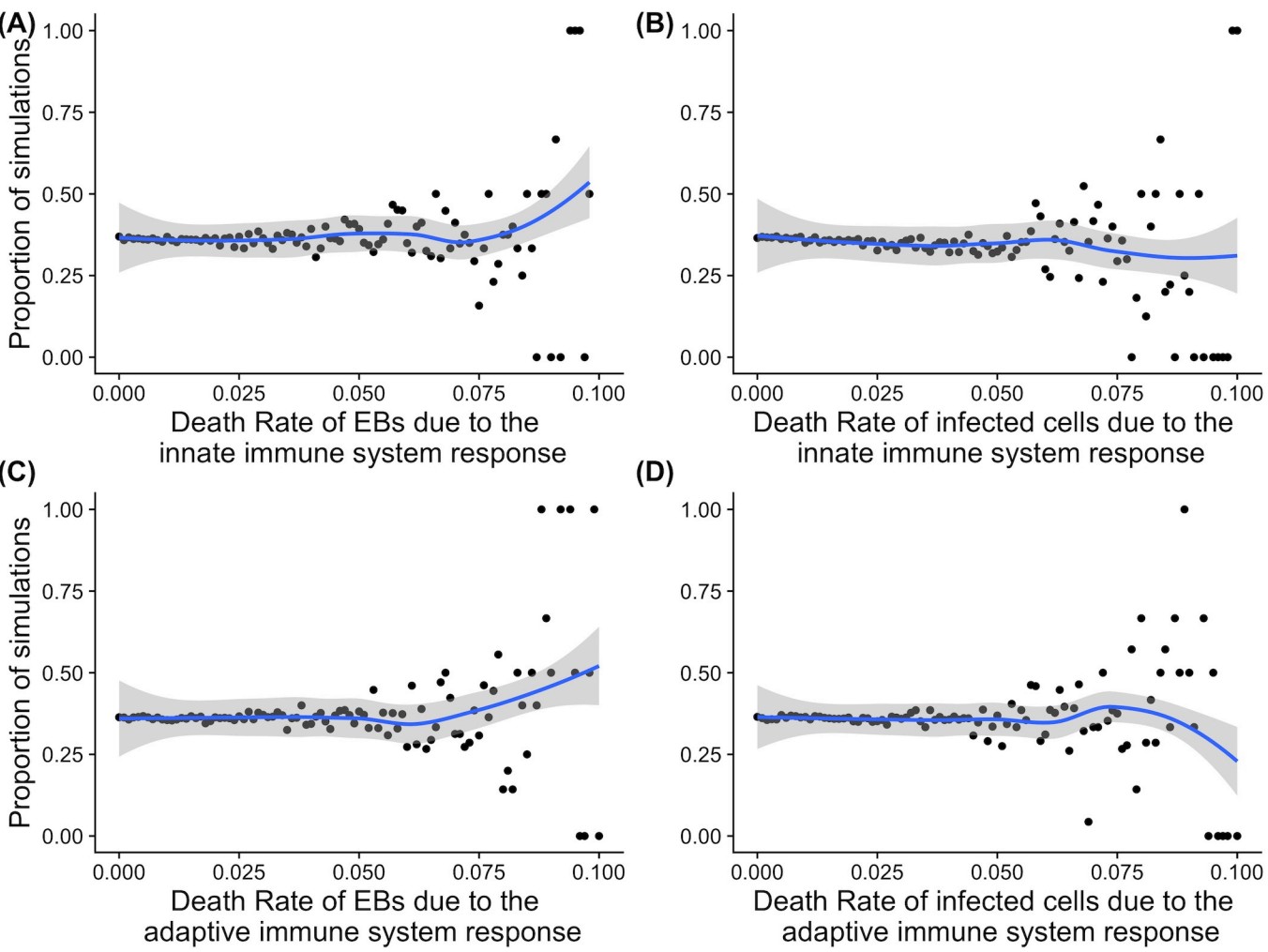

**Fig 7. Graphical representation of the impact of the immune response and subsequent chlamydial death rate on the proportion of simulations that result in ascension.** The proportion of simulations where the infection ascends was computed (total 252,500 simulations represented in each figure). Each figure shows the proportion of simulations where the infection ascended on the y axis and on the x axis the death rate of EBs and infected cells due to different components of the immune system (represented as broad scales of response). (A) Impact of EB death rate due to the innate immune system on the proportion ascending, (B) Impact of infected cells death rate due to the innate immune system on the proportion ascending, (C) Impact of EB death rate due to the adaptive immune system on the proportion ascending, and (D) Impact of infected cells death rate due to the adaptive immune system on the proportion ascending. These are simulated over a range of infectious loads (10 simulations at each value of infectious load from 1 to 10001 in steps of 100.The solid blue line indicates a loess fit to the proportion of simulations the ascended and the shaded area represents 95% Confidence Intervals of the loess regression. Dots represent simulations at a particular parameter value from a total of 252, 500.

responses in the movement of the pathogen, in spite of its role in controlling infections, and thus the findings here from our modelling are consistent with the literature on neutrophils. Our model assumes that the neutrophil response is critical in epithelium to the cervical canal, and so the neutrophil response plays a different role in modulating ascension to that observed from the global innate immune system, in regards to the relationship between high and low immune response and ascension [8]. It is not clear from the available literature if the neutrophil response varies independently between hosts, or if it occurs as a function of the amount of *C. trachomatis* present at a particular site in the epithelium (or both). From the perspective of the pathogen, one could view the evolution of *C. trachomatis* alongside the neutrophil response as a process that has selected for these adaptations to exploit this natural defence

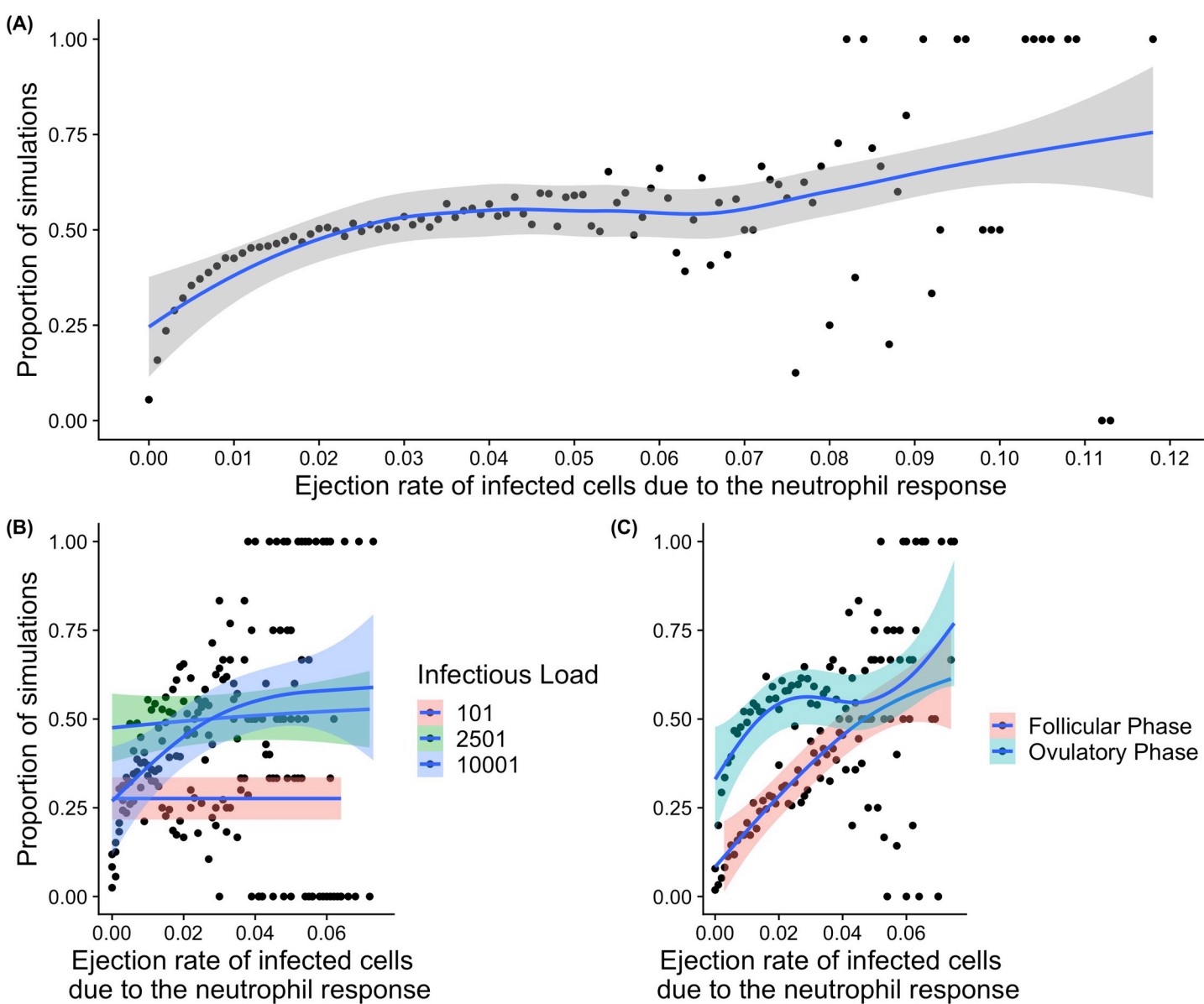

**Fig 8. Graphical representation of the proportion of infections that ascend in the simulations from the model and the role of the neutrophil response.** The figure shows the proportions of simulations (total of 252, 200 conducted for each instance) where infection was predicted to ascend (y axis). The ejection rate (movement to the cervix from the epithelium) of infected cells related to the neutrophil response (x axis). (A) Ascension and the ejection rate due to the neutrophil response. (B) Impact of initial infectious load (pink represents a low infectious load of 101, green a moderate infectious load of 2501, and blue 10,001). (C) Time of infection relative to the menstrual cycle (follicular phase represented by pink, and ovulatory phase represented by blue). These are simulated over a range of infectious loads (10 simulations at each value of infectious load from 1 to 10001 in steps of 100. In each graph the solid blue line indicates a loess fit to proportion of simulations resulting in ascension, where the shaded area shows is the 95% confidence interval. Each dots are represents simulations at a particular parameter value, from a total of 252,500.

mechanism to ensure pathogen survival. The evidence we provide here suggests that an influx of neutrophils may be a useful predictor of an ascending infection.

This interaction between the host's defence mechanism and the infection must be time-dependent for an infection to successfully ascend according to the outcomes shown here. As migration from the initial site of infection during the earlier phases of the menstrual cycle will more likely spread the infection too thin, with insufficient material left in the epithelium in

order to ensure to short term survival of the infection and to ensure adequate infectious material for successful transmission to new hosts. Our simulations indicate that both the infecting load and the menstrual cycle moderate ascension, and that this moderation occurs in a dynamic and dependent way. There is limited biological data to compare with the proportions of infections that ascend that we have predicted here in the model. Ascension of chlamydia is a necessary, but not sufficient condition for pathology (such as PID) to develop, and we know that PID occurs in approximately 9.5% of chlamydial infected women [1]. In a majority of our simulations where ascension did occur, the amount of material that ascended was relatively small (although 9% of ascensions in one model had greater than 1000 organisms, Fig 5). A case-control study of women with positive (cases) and negative (controls) tests for chlamydia, gonorrhoea or bacterial vaginosis, found the women in the follicular phase of their menstrual cycle had an adjusted odds ratio of 4.5 (95% CI (1.6–12.4) of experiencing upper genital tract infection [27]. Although the timing of biopsy in this clinical evidence is not necessarily equivalent to the timing of ascension, this study supports the findings we present that menstrual cycle moderates ascension through peristalsis. Our modelling finds that infections that are initiated in this stage of the menstrual cycle are least at risk of ascending.

A key assumption of this modelling exercise is that uterine peristalsis changes continuously with the menstrual cycle. For an infection to ascend, it requires exact timing with the menstrual cycle in order to utilise the peristaltic activity of the uterus. However, women with dysfunctions of the peristaltic mechanisms, such as hyperperistalsis, may experience peristaltic activity at three times the rate of women without these dysfunctions [28]. Further, there is some evidence that women with high uterine peristalsis have reduced pregnancy rates due to the presence of intramural fibroids [29]. Although speculation, it is plausible the women with increased peristaltic activity due to uterine fibroids would be at increased risk of ascending infections.

The limitations of this work should be noted. This exercise characterises the process through simulated realisations of the model, and it is not calibrated or fit against biological datasets. Additionally, as with all modelling, our results rest on a set of assumptions that can potentially be modified to realise different outcomes. This cannot be avoided, although we have been as rigorous as possible in simulating over a broad range of parameter values and including stochastic terms to ensure the dynamic nature of the observations is encompassed. Unlike previous mathematical models of chlamydia [13, 18, 22, 23], the model presented in this study incorporates stochastic elements that samples across a broad range of parameter values, which influences realised outcomes of the model and provides some robustness to the results presented [21]. Also, we do not continue the simulations beyond the point of upwards peristalsis within the stage of the cycle, meaning the longer term implications for the longer time course of the infection have not been considered in the model. The results may be extrapolated to determine the chances in subsequent cycles after the initial cycle, although a model specifically developed to answer the likelihood of ascension in subsequent months may be a more appropriate approach for longer range estimates. A significant component of chlamydial function not included in the model is the differing strategies that *C. trachomatis* uses to exit the cell. The mechanisms that govern lysis and extrusion cellular release pathways are distinct and may vary in the rate at which they occur and the amount of *C. trachomatis* released from the cell [20]. Extrusions of chlamydial inclusions have the ability to utilise macrophages as intracellular survival niches [30]. The pathogen may then be able to exploit macrophages as a vehicle of dissemination within a host. The macrophage response has implications for other bacterial pathogens, in particular *Neisseria gonorrhoeae*, which has the ability to inhibit apoptosis and use macrophages as a replicative survival niche [31, 32].

In conclusion, we have proposed a process for the ascension of chlamydial infections, by considering the available clinical evidence and experimental observations of the pathogen and combining these observations into a stochastic model that allows us to understand the influential factors of ascension. The dynamics of the process are complex, nonlinear, and stochastic, so we model this in order to reason about the moderating factors. From our simulation results, we propose that cervical neutrophil responses and infectious burden could be useful as early indicators of the likelihood of pathology. This would need to be validated with further examination of neutrophil levels as predictive markers for pathology development of women who have recent exposure to *C. trachomatis*. Further we propose that the model could be more broadly applied to other STI to identify relevant factors for ascension and pathology risk in the cases of those pathogens.

## Supporting information

**S1 Fig. A histogram of the amount of ascended bacteria under different distributional assumptions.** Each histogram represents 12,625 simulations from the model, where the underlying distribution for the rate parameters has been selected from an exponential distribution, a gamma distribution and a log-normal distribution. All distributions have identical means of 1/100.
(TIF)

**S1 Text. Provides information on the Parametrisation of key distributions.**
(DOCX)

## Author Contributions

**Conceptualization:** Wilhelmina May Huston.

**Data curation:** Torrington Callan.

**Formal analysis:** Stephen Woodcock, Wilhelmina May Huston.

**Investigation:** Torrington Callan, Stephen Woodcock, Wilhelmina May Huston.

**Methodology:** Torrington Callan, Stephen Woodcock.

**Writing – original draft:** Torrington Callan.

**Writing – review & editing:** Stephen Woodcock, Wilhelmina May Huston.

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
