## [Decision Letter · Decision Letter 0]

18 Apr 2021

Dear Dr. Huston,

Thank you very much for submitting your manuscript "Ascension of Chlamydia is moderated by uterine peristalsis and the neutrophil response to infection" for consideration at PLOS Computational Biology.

As with all papers reviewed by the journal, your manuscript was reviewed by members of the editorial board and by several independent reviewers. In light of the reviews (below this email), we would like to invite the resubmission of a significantly-revised version that takes into account the reviewers' comments.

We cannot make any decision about publication until we have seen the revised manuscript and your response to the reviewers' comments. Your revised manuscript is also likely to be sent to reviewers for further evaluation.

Sincerely,

Janneke Heijne, Ph.D.

Guest Editor

PLOS Computational Biology

Rob De Boer

Deputy Editor

PLOS Computational Biology

Reviewer's Responses to Questions

**Comments to the Authors:**

Reviewer #1: This is an interesting and well written paper on several factors that might moderate ascension of Chlamydia trachomatis infections to the upper genital tract in women. Results of the study indicate that infectious load, menstrual cycle timing, and the neutrophil response are important factors in the ascension of chlamydia. Results might increase insight in the pathology of chlamydia infections and factors might enable specific testing for at-risk of high risk women, or aid the development of interventions at timing of infection. In light of the current paradigm of chlamydia control and the question “what should we do with chlamydia control?” this paper is relevant and contributes to finding new ways of identifying high risk women.

Specific comments/questions include:

1. Line 67: In the introduction, several factors were mentioned to moderate in the ascension of chlamydia (menstrual cycle, stage of infection, immune status, genotype factors (host and pathogen) and infectious load). But not all were elucidated further in the introduction (for example genotype factors), while all factors were included in the model.

2. Line 114: “The PMN response was observed to result in detached superficial epithelia from the mucosa, including infected cells and elementary bodies that remained healthy and intact during the process. It was suggested by Rank et al. that this response was a defensive mechanism from the perspective of the host organism in that it allows for infected cells that have been ejected from the epithelium to be cleared from the infection site. Further, it was proposed that from the perspective of the Chlamydia, this process allowed for movement away from infected sites where there is a lack of susceptible uninfected cells, transporting to new infection sites via fluid dynamics or tissue movement.” Wouldn’t you expect that, primarily the fluid dynamics or vaginal secretion, will descend the infected cells rather than ascend? How does that interreact?

3. Line 144: “Elucidating the processes involved in chlamydial ascension will provide a better understanding of the pathogenesis of Chlamydia associated PID and infertility. In turn, the moderating factors could enable more specific testing for at-risk women, or development of other interventions when infections are diagnosed.” Although I agree that elucidating processes around the ascension of chlamydia is relevant, specific testing for at-risk women concerning the menstrual cycle requires the need of timing of infection. Even more so the timing of ascending of the bacteria. How long might an infection reside in the lower genital tract before ascending?

4. Line 182: Is it correct that you assume that ejected infected cells contain viable RBs that will differentiate back in EBs? In other words that neutrophils were unable to inactivate chlamydia?

5. Line 348: “However, increasing the rate at which extracellular bacteria infect target cells does appear to increase the chances of ascension. This points towards the increased utility of infected cells to the infection, as they influence migration and will produce greater amounts of bacteria in the future.” How would the rate at which EB infect target cells be increased? Can you elaborate more on how this points towards the increased utility of infected cells?

6. Line 372: “The relationship between infectious load and the menstrual cycle further complicates matters. Infections with smaller infectious loads experience linearly decreasing chances of ascension as the menstrual cycle progresses, so an infection that occurs later in the cycle may have reduced chances of ascending. However, the opposite applies to high infectious loads, where infections have a higher chance of ascending as the menstrual cycle progresses closer to peristaltic movement in the cervico-fundal direction.” It seems that the chance of an ascending infection increases when you have either low initial loads early in the menstrual cycle or high loads later in the menstrual cycle. Are these chances comparable? Or is the chance of ascending with higher loads but later in the menstrual cycles still significantly lower compared to low initial loads early in the menstrual cycle. The conclusion in the abstract would suggest the latter.

7. Furthermore the interaction between infectious load and menstrual cycle: Is that based on the menstrual cycle during time of initial infection? “Infections with smaller infectious loads experience linearly decreasing chances of ascension as the menstrual cycle progresses”, but if the infection is not cleared it might produce enough material so that in the following menstrual cycle there would be enough material that might cause a new infection in the upper genital tract? For example after one year after initial infection, without treatment, only 50% of women have cleared the infection. Could you elaborate on that?

8. I’m am somewhat confused about how the neutrophil response relates to high and low initial infectious loads. In the introduction it was said that in case of high infection loads the immune response was sufficient to prevent ascension but not for lower loads. But in the results it was found that infections with high neutrophil responses were more likely to be one that ascend. Could you elaborate on that?

9. Line 411: You propose to investigate the strict set of markers to investigate in a cohort study to examine development of upper genital tract pathology. How would you set up such a cohort?

10. Line 477: A study from Korn concluded that he proliferative phase of the menstrual cycle seems to be the primary risk factor for ascending infection by organisms associated with pelvic inflammatory disease. From the results of the study I don’t think that it can be concluded that timing of biopsy is equal to timing of ascension. The ascension might have happened days or weeks before and might differ between hosts.

Reviewer #2: Manuscript Number: PCOMPBIOL-D-20-02253

Title: Ascension of chlamydia is moderated by uterine peristalsis and the neutrophil response to infection

In this study, the authors constructed a stochastic model that describes the dynamics between extracellular chlamydial elementary bodies and infected cells containing chlamydial reticulate bodies in the epithelium and the probability that these ascend to the cervical canal. It is investigated how factors such as the infectious load, the menstrual cycle timing, and the neutrophil response could affect chlamydial ascension in women.

This is a very interesting study. Although it is clear that ascension can occur, it is unclear which factors facilitate or hinder ascension. The authors have developed a simple model that captures the main elements of this process, thus illustrating the role of some important factors, such as the infectious load or the phase of the menstrual cycle.

However, the study needs quite some improvement. The notation in the mathematical formulas is inconsistent and unclear and methods need some clarifications. English language needs improvement. References have to be added throughout the manuscript to support the arguments. The discussion contains too many repetitions, but does not cover important items, such as limitations and discussion/comparisons with similar studies. These points are explained in detail below.

My main consideration about the study is that, as the authors mention in the discussion (page 14), there are hardly any data to compare with model results for some level of validation. That makes the study kind of speculative or hypothetical. Also, the authors provide no evidence/reference to support their assumptions for the distributions of the parameters (e.g. Gamma, Beta, Negative Binomial and their means/shape parameters). If two or three of these parameters have a different distribution or the shape parameter is half of what the authors assumed, the results could be quite different. Although this is inherent for all models (input quality determines output quality), modelling studies nowadays provide extensive support for the model parameters (from earlier studies or directly estimated from data) AND model validation or fitting to data. That provides some level of confidence in the qualitative conclusions of modelling studies, which here lacks.

Some additional points that could improve the study are listed below.

1. Line 187: In discussion, please explain how the results are affected by the assumption of instantaneous burst as opposed to continuous.

2. Lines 215-223: The formula’s presented here are incomprehensible, especially since the notation in Figure 2 seems to imply that the rate is I*k1 or M1 * YC1 etc. The information in these formula’s is more or less repeated in the paragraphs from line 237 to 315. The authors could reorganize the text by deleting the repetitions and provide all information about each process in the model together (e.g. all formulas, explanations, assumptions, distributions, parameters about the immune response; then those about inclusion and bursts etc).

3. Pages 8-9: Please remove Latex notation and use the same notation for each variable/parameter (e.g. gamma1 or gamma_1 in lines 267-270; or \\gamma and Greek gamma in lines 307,308).

4. Lines 270-271: Gamma(1,00): what is after the comma?

5. Pages 8-9: Please provide some references or explanation for the assumptions about the distributions and the values of the parameters of the distributions used.

6. Lines 311-313: Ascension is defined as a yes/no event, depending on whether ascension occurred “when the simulation time finishes”. Maybe I have missed it, but when does the simulation finishes? How does the % ascending depend on the length of this time? How would the results about the proportion ascending differ with twice as long or half of this time?

7. Legends of Figures 3-8: The legends discuss the results shown in the plots and draw conclusions. This is NOT the purpose of the legends. Please remove all text describing results and conclusions. The legends should explain what is shown in the plots and the scenarios, parameters, or methodology for each specific plot or result shown. Also, please add units where appropriate (e.g. “time” and “load” are measured in what?)

8. Lines 323-332: These are not results of this study; please remove this paragraph.

9. Lines 334-341: Please include reference to figure 3 at the beginning of the paragraph. The reader has to read the whole paragraph to understand where these results are shown. What is scenario A, B, C? do you mean plots A,B,C? What is the difference between the three plots? Are they just 3 realizations (with same parameter distributions and same initial conditions? If not, how do they differ? If they are with same parameters and initial conditions, then please add explicitly something like “plots A, B, C show three realizations of the same stochastic process with same parameter values and initial conditions showing the three major types of the course of the infection: not ascending, ejecting but finally dying out, surviving”.

10. Results: The whole section provides mostly a qualitative description of the results, e.g. this increases when that rates increase etc. Please provide some indicative quantitative results. For example, if the rate is 0.05, then the % ascending is 25%; with a rate of 0.10, the % ascending is 35% etc. Also, please remove conclusions, like “this points towards the increased utility of…”, and place them in the discussion.

11. Legend figure 4: “(A) increasing the rates of burst does not appear to impact the chances of ascension”: Again, this is description of result and drawing conclusions. The legend of this plot should be something like “The proportion ascending with different rates of burst”. Please make similar changes to all legends.

12. Lines 356-358: This text is unclear. Please rephrase.

13. Legend Figure 5: This plot is not a “summary of simulations”. Please use a proper name to describe this kind of figure. Please mention the number of simulations used for this plot and delete all conclusions.

14. Legend Figure 6: The horizontal axis shows the “amount of time since infection”; please add units and change the legend since the phases of the cycle are not shown. The authors could add the phases.

15. Figure 7: Please explain/comment on the fact that the percentage of simulations ascending with high rates varies from zero to 100%! The legend for A reads “Innate immune system impact on EB death rate”, but I think it shows the impact of EB death rate due to innate immune system on % ascending”. The same for the other plots. Please update.

16. Discussion: This section is quite long and contains a lot of repetition from introduction and results. Please remove repetitions. Please add a discussion of limitations of the study (e.g. impact of assumed distributions and parameters of the distributions), as well as comparison with earlier studies on the subject.

**Have the authors made all data and (if applicable) computational code underlying the findings in their manuscript fully available?**

Reviewer #2: **No: **No data or references are given to support distributions and values of model parameters

PLOS authors have the option to publish the peer review history of their article (what does this mean?). If published, this will include your full peer review and any attached files.

Reviewer #1: No

Reviewer #2: No

**Have all data underlying the figures and results presented in the manuscript been provided?**

Reviewer #1: Yes
---

## [Decision Letter · Decision Letter 1]

14 Jul 2021

Dear Dr. Huston,

Thank you very much for submitting your manuscript "Ascension of Chlamydia is moderated by uterine peristalsis and the neutrophil response to infection" for consideration at PLOS Computational Biology. As with all papers reviewed by the journal, your manuscript was reviewed by members of the editorial board and by several independent reviewers. The reviewers appreciated the attention to an important topic. Based on the reviews, we are likely to accept this manuscript for publication, providing that you modify the manuscript according to the review recommendations.

Sincerely,

Janneke Heijne, Ph.D.

Guest Editor

PLOS Computational Biology

Rob De Boer

Deputy Editor

PLOS Computational Biology

[LINK]

Reviewer's Responses to Questions

**Comments to the Authors:**

Reviewer #1: Following the first peer-review round, the authors made significant improvements to the manuscript. Elaborations and explanations have been made and the manuscript is more concise, specifically the discussion section.

Specific comments:

1. Table 1 is an improvement and gives insight in the parameters. The table itself is rather large, which decreases readability. Could you make the table more concise (for example Colom wide modifications).

2. Headings in the result section come across as conclusions, for example line 393 and 413-414.

3. The discussion starts off with a limitation and comparison of the literature on line 443-451; commenting on dysfunctions of the peristaltic mechanisms, this comes out of nowhere. It might be worthwhile to mention it somewhere in the discussion, but please start the discussion focusing on the results of studied factors that influence chlamydial ascension.

4. Line 525 please choose either examining/evaluating.

5. In the conclusion you mentioned you could consider to further examining neutrophil levels as predictive markers for pathology development, but you also propose to that neutrophil levels could be useful as early indicators for pathology likelihood. The way that this is phrased weakens the conclusion. What is it you want to conclude? Could neutrophil levels potentially be useful as early indicators and therefor further examination is needed? Or is it useful even without further examination?

Reviewer #2: PCOMPBIOL-D-20-02253R1

The manuscript is considerably improved and the authors have addressed most of my questions. However, there are some points that need some clarification/improvement:

1. How much time were the simulations carried out to determine whether ascension occurred or not? Did the authors examine/assess whether the % ascending depends on the duration of simulations? I had raised this point in my first review (point 6), but the response of the authors was unclear: they write that the “simulation is stopped when peristalsis causes material to begin ascending”. In some simulations, ascension does not occur, so my question is: how long were these simulations carried out? Did the authors assume that all infections end with either 0 bacterial load or ascension and continued the simulations until one of the two events occurred? In that case, what was the max time needed to reach one of the two endpoints? That helps to understand the time scales. Please add this information and explanation in the main text.

2. A point relating to time: in the author’s response to point 7 of Reviewer 1, the authors state that “an infection in month 2 of 1000 EBs is equivalent to a new infection of 1000 EBs”. However, an infection of 1000 EBs in month X, may become an infection of 2000 EBs (for example) in month Y – and that possibly depends on the time Y-X? Can you explain how that was modelled? If infections in subsequent menstrual cycles are equivalent to new infections, how was the menstrual cycle included in the model?

3. Lines 233-239: “chlamydia” usually refers to the disease itself. Please use here another term instead of chlamydia and, preferably, the same term you use in the previous paragraphs (EB, if I understand it correctly) to avoid confusion.

4. Fig.3: In line 372, it says that the three scenarios A, B, C show realizations of the same stochastic process with distinct parameter values and same initial conditions. In the legend and in the Results section, it is stated that the three scenarios A, B, C, start with different initial load. I would expect that the “initial load” is one of the initial conditions, not a parameter. What exactly is the initial load? Is it the value of one of the model variables at t=0 or one of the parameters and which one? Please give the actual initial load (not an approximation, like “~800 EBs”; and not only a comparison “more/less than in A”) for each of the scenarios A,B,C.

5. The description of results from Fig.3 seems to imply the importance of the “initial load” for the outcome of ascension. That raises the following question: what was the initial load in the other results (Fig. 4-8) and how were these results affected by the initial load?

6. Lines 397-398: It is nice that the authors added these percentages in the text. It is a pity, however, that these cannot be “seen” in the plot. Could the authors add more labels on the axes (e.g. showing values between 0 and 0.05 on the horizontal axes and values between 0 and 0.25 on the vertical axes) to increase readability of the plots.

7. The authors use various terms for the disease chlamydia and the organism (C. trachomatis) causing the disease and sometimes use the same term for the disease and the organism. To avoid confusion, it is advisable to use distinct terms for the organism and the disease and to use these terms consistently throughout the manuscript.

**Have the authors made all data and (if applicable) computational code underlying the findings in their manuscript fully available?**

Reviewer #1: None

Reviewer #2: **No: **

PLOS authors have the option to publish the peer review history of their article (what does this mean?). If published, this will include your full peer review and any attached files.

Reviewer #1: No

Reviewer #2: No

Figure Files:

Data Requirements:

Reproducibility:

References:

---

## [Editor Report · Decision Letter 2]

19 Aug 2021

Dear Dr. Huston,

We are pleased to inform you that your manuscript 'Ascension of Chlamydia is moderated by uterine peristalsis and the neutrophil response to infection' has been provisionally accepted for publication in PLOS Computational Biology.

Best regards,

Janneke Heijne, Ph.D.

Guest Editor

PLOS Computational Biology

Rob De Boer

Deputy Editor

PLOS Computational Biology

---

## [Editor Report · Acceptance letter]

1 Sep 2021

PCOMPBIOL-D-20-02253R2 

Ascension of Chlamydia is moderated by uterine peristalsis and the neutrophil response to infection

Dear Dr Huston,

I am pleased to inform you that your manuscript has been formally accepted for publication in PLOS Computational Biology. Your manuscript is now with our production department and you will be notified of the publication date in due course.

With kind regards,

Katalin Szabo
